# Predictive Glycaemic Response of Pasta Enriched with Juice, Puree, and Pomace from Red Cabbage and Spinach

**DOI:** 10.3390/nu14214575

**Published:** 2022-10-31

**Authors:** Jinghong Wang, Margaret Anne Brennan, Charles Stephen Brennan, Luca Serventi

**Affiliations:** 1Faculty of Agriculture and Life Sciences, Lincoln University, P.O. Box 85084, Christchurch 7647, New Zealand; 2Riddet Research Institute, Private Bag 11 222, Palmerston North 4442, New Zealand; 3School of Science, RMIT, Melbourne, VIC 3000, Australia

**Keywords:** glycaemic index, vegetable pasta, α-amylase inhibition, dietary fibre, antioxidant

## Abstract

This study reports the digestibility and nutritional quality of pasta made from durum wheat semolina which was partially substituted by puree, juice or pomace from spinach and red cabbage. The results show that 10% substitution of semolina with red cabbage pomace and spinach pomace, 1% substitution of spinach juice, and 2% substitution of spinach puree significantly reduced the area under the curve of the in vitro starch digestion. This reduction was due to a combined effect of decreased starch content, increased dietary fibre content and inhibition of α-amylase caused by vegetable material addition. Total phenolic content (TPC) and antioxidant capacity increased significantly on raw, cooked and digested samples of vegetable fortified pasta compared to control. The β-carotene content of spinach pasta (raw, cooked, and digested) was also higher than that of control. At the 1% substitution level, the juice was more efficient in improving the antioxidant capacity of resultant pasta compared to puree or pomace.

## 1. Introduction

There is great interest in producing health-promoting foods fortified with plant bioactive ingredients [1]. Pasta is a staple cereal food worldwide that is widely accepted and thus could be a good food system to incorporate healthy ingredients [2]. The glycemic index (GI) of pasta is lower than other staple foods such as bread and rice [3]. Papoutsis et al. [4] found that some phytochemicals, such as polyphenols saponins, and proteins from vegetables could deliver α-amylase and α-glucosidase inhibition effects. Phenolic compounds from fruit have been shown to affect starch degradation and reduce the potential glycaemic impact of starchy foods [5]. A high level of dietary fibre has also been reported to attenuate starch hydrolysis and glycaemic response [6]. Hence, adding vegetable ingredients to pasta may be one option to produce low GI pasta.

Vegetables contain many health-promoting bioactives that traditional pasta lacks [7]. Some of those compounds such as dietary fibre, proteins, and polyphenols may be useful to improve the nutritional value of pasta. Substitution of semolina with vegetable powder is typically used to produce vegetable pasta to enhance its nutritional value [8,9]. Other authors utilised vegetable juice or by-product (such as skin pomace) to increase pasta’s nutrition [1,8,10,11,12]. The use of these by-products in foods have consumer acceptability issues, but consumers generally perceive them to be advantageous from a nutritional basis [13].

However, limited research has been carried out to compare different forms of vegetables regarding adding them to pasta formula and comparing the nutritional value of resultant pasta. Therefore, pasta enriched with other forms of vegetables, such as puree, juice, and pomace, was investigated in this study. The aim is to determine how different forms of vegetables influence the glycaemic response, antioxidant capacity, and other significant nutrients (including total phenolic compounds and β-carotene). Two types of leaf vegetables (spinach, red cabbage) were selected. Spinach was reported to be an α-amylase inhibitor, and its inhibition effect can compete with acarbose, a known antidiabetic agent [14]. Spinach is also rich in lipid-soluble nutrients such as lutein and β-carotene [14]. Red cabbage was found to be rich in water-soluble phytochemicals that have strong antioxidant capacity [4]. It should be noted that the addition of juice and puree to pasta have limitations due to the techno-functional properties of the pasta such as cooking quality and consumer appearance. Juice contains low solid content (typically 5–15%) and this can dilute the solid components of pasta and result in a weaker pasta matrix being formed [15]. Thus achieving a high substitution level based on dry matter may be impossible when using juice or puree due to excessive hydration. The preliminary study (Appendix A) found that vegetable powder showed no significant difference compared to vegetable puree when added to pasta in key technical tests such as elasticity, firmness, and cooking loss. However, the antioxidant ability of powder-enriched pasta was lower than puree-enriched one, hence the puree was used instead.

## 2. Materials and Methods

### 2.1. Materials

Semolina (Sun Valley Foods Ltd., Christchurch, New Zealand), fresh spinach and red cabbage were bought from the local supermarket (New World Supermarket., Lincoln, New Zealand).

### 2.2. Vegetable Preparation

Spinach and red cabbage were washed thoroughly, and their roots were removed. The stem and leaf were put into a juicer (Model: Oscar Neo DA 1000; ATURE’S WONDERLAND Ltd., Australia), and the pomace and juice were collected separately. The vegetable juice was placed into a glass jar with a cap and stored at −18 °C until use. The pomace was spread in a tray and put into an oven to dried at 60 °C for 7 h. The dried pomace was then ground to a powder using a coffee grinder (Model: Breville BCG200; Breville Pty Ltd., Sydney, Australia) for 10 s twice, and the resultant pomace powder was stored in a ziplock plactic bag at room temperature. The spinach puree and red cabbage puree were produced by freshly mixing juice and pomace together in a blender (Nutri-bullet NBO7200-121-DG; Capitalbrands Ltd., Boston, MA, USA). The purees were collected in a glass jar with a cap and stored at −18 °C. Before use, the puree was defrosted at room temperature for 2 h and put into the blender again to homogenise.

### 2.3. Pasta Preparation

Pasta was prepared using a lab-scale pasta machine (Model: MPF15N235M; Firmer, Ravenna, Italy) with spaghetti die of 2.25 mm diameter. The vegetable pomace fortified pasta was prepared by mixing the pomace with semolina in the pasta machine, and then 40 °C water was added and then mixed for 20 min before extruding to pasta. The formula is shown in Table 1. The puree and juice were defrosted and warmed to 40 °C in a water bath. Then, either puree or juice and 40 °C water was added to semolina in the pasta machine to extrude to pasta. The substitution level of juice and puree pasta was based on dry matter, according to the solid content measurement of the raw material.

### 2.4. Total Starch and Dietary Fibre Analysis

Megazyme starch analysis kits (Megazyme International Ireland Ltd., Wicklow, Ireland) were used to determine the total starch content of vegetable pasta according to the AOAC Official Method 966.11. The soluble dietary fibre (SDF), insoluble dietary fibre (IDF), and total dietary fibre (TDF) of freeze-dried cooked pasta was measured in duplicate by dietary fibre assay kit (Megazyme International Ireland Ltd., Wicklow, Ireland) using the AOAC official method [16].

### 2.5. In Vitro Digestion

An in vitro digestion was carried out according to Peressini et al. [17] with slight modification. Frozen pasta (5 g) was defrosted for 120 min at room temperature and cooked to OCT in boiling tap water (250 mL). The pasta was then drained for 1 min and cut into 2 mm strands. The pasta (2.5 g) was weighed in a 60 mL plastic biopsy pot that was placed on a pre-heated magnetic heat stirring block (IKAAG RT 15, IKA-WERKE Gmbit & Co., Staufen, Germany). The sample in the biopsy pot was then stirred with 30 mL distilled water and 0.8 mL 1 M HCl at 37 °C. Gastric digestion was mimicked by adding 1 mL 10% pepsin solution and stirring for 30 min. Then, 2 mL 1 M NaHCO_3_ and 5 mL 0.1 M Sodium maleate buffer (pH 6) were added. A 1 mL aliquot was taken (0 min), and 5 mL 2.5% pancreatin was then added. Further 1 mL aliquots were taken at 30 min, 60 min, and 120 min during the digestion. The reducing sugar was measured using 3,5-nitrosalicylic acid (DNS). The rest digesta was collected and stored at −18 °C for antioxidant analysis and β-carotene analysis.

### 2.6. Sample Extraction for Antioxidant Analysis and α-Amylase Inhibition

Raw pasta and cooked pasta samples were freeze-dried. 2.5 g of freeze-dried pasta or 2.5 mL digesta were extracted by stirring overnight at 20 °C in 25 mL of 70% methanol solution. The mixtures were then centrifuged at 2500 rpm for 10 min. The supernatant was collected and kept at −18 °C until analysis.

### 2.7. Antioxidant Analysis

Total phenolic content (TPC) was measured using 0.2 N Folin–Ciocalteu reagent (Sigma, St Louis, MO, USA) according to Lu et al. [18]. Ferric Reducing/Antioxidant Power (FRAP) was tested by a working reagent (mixture of 300 µM acetate buffer, 10 mM TPTZ solution and 20 mM FeCl_3_ at 10:1:1 (v/v/v)) based on the method from Rachman, A. Brennan, Morton and Brennan [2]. ABTS (2,2′-azino-bis(3-ethylbenzothiazoline-6-sulfonic acid)) Radical Scavenging Capacity was determined following Rachman, A. Brennan, Morton and Brennan [2] using ABTS working solution.

### 2.8. Determination of α-Amylase Inhibition

The α-amylase inhibition activity of the extract was determined according to Sultana et al. [19]. A 0.1 mL sample extract (From 2.6) was dissolved in sodium phosphate buffer (0.02 M Na_2_HPO_4_ and 0.02 M NaH_2_PO_4_ pH 6.9 with 0.006 M NaCl). A 0.1 mL of the sample dilution was then mixed with 0.1 mL of pancreatic α-amylase solution (3000 U, Megazyme International Ireland Ltd., Wicklow, Ireland). The mixture was then incubated at 37 °C for 10 min. A 0.1 mL of 10 mg/mL starch solution (dissolved in the same buffer) was then added to terminate the reaction, and the mixture was incubated at 100 °C for 30 min. The test tube (with the mixture) was then cooled down in a water bath and 2.7 mL of distilled water was then added to the test tube. After mixing, the absorbance was read at 540 nm. The phosphate buffer was used as blank, and acarbose was used as the positive control. The results are presented as IC_50_ (mg/mL).

### 2.9. Sample Extraction for β-Carotene Analysis

The aqueous phase of digesta was isolated by centrifugation (3000× RPM) for 10 min. 6 mL of aqueous phase digesta, or 2 g of freeze-dried raw and cooked pasta, or 2 g of spinach raw material (spinach pomace, freeze-dried spinach puree and juice) were extracted with 4 mL of methanol: water: acetone (2:1:1) and vortex for 1 min. The upper acetone phase was transferred to a centrifuge tube and 2 mL acetone was added to the previous sample tube repeat extract twice. 2 mL of hexane was then added to the centrifuge tube and vortex for 1 min, following a centrifuge (3000× RPM) for 2 min. The hexane extraction was repeated two more times, and the supernatant was collected and rotary-evaporated at 30 °C. The dried extract was mixed with 2 mL of methanol and filtered through a 0.45 µm filter for HPLC analysis.

### 2.10. HPLC Analysis of β-Carotene

Agilent 1100 series (Agilent Technologies, Walbronn, Germany), was used to analyze β-carotene. It equipped with a binary pump, auto-sampler with thermostat, kept the temperature at 25 °C in a column oven compartment. Chromatographic analysis was performed using a column (EXL-1110-1546U, ACE 3µ C18-PFP 150 mm × 4.6 mm, Advanced Chromatography Technologies, Aberdeen, Scotland). HPLC conditions were as follows: solvent A (Methanol 100%), solvent C (Ethyl-acetate 100%), separation of β-carotene achieved by a step gradient shown in Appendix A with a flow rate 0.8 mL/min based on van Leeuwe et al. [20]. Qualification was carried out by an external standard (Sigma C4582). The results were calculated by duplicated test.

### 2.11. Statistical Analysis

All determinations were carried out in triplicate except where otherwise stated. ONE WAY ANOVA was evaluated by the Duncan test (SPSS version 16). Correlation tests were performed by two-tailed Pearson tests to analysis significant correlations at *p* ≤ 0.05, and *p* ≤ 0.01 (SPSS version 16).

## 3. Results and Discussion

### 3.1. In Vitro Starch Digestion Versus Total Starch Content

The amount of reducing sugars released over 120 min digestion is shown in Figure 1a,b. Reducing sugars reached a peak value at either 20 min or 60 min for all pasta samples. This observation was similar to the previous study conducted by Lu, Brennan, Serventi, Liu, Guan and Brennan [18]. Figure 1a shows that significantly less reducing sugar was released from spinach juice 1% pasta and spinach pomace 10% pasta compared to control puree semolina pasta. However, only red cabbage pomace 10% pasta released significantly less reducing sugar than control. Figure 1c shows the standardised incremental area under the curve (AUC) of spinach pasta and red cabbage pasta compared to the control. Spinach juice 1%, spinach puree 2%, and spinach pomace 10% pasta show a significantly lower (*p* < 0.05) AUC compared to control. Red cabbage pomace 10% pasta presents a significantly lower (*p* < 0.05) AUC while cabbage materials replaced 1% and 2% semolina (red cabbage juice 1%, red cabbage puree 1%, red cabbage puree 2%, red cabbage pomace 1% and red cabbage pomace 2% pasta) has not reduced the AUC. It may indicate that for red cabbage material, a high substitution level may be required to achieve lower AUC. Cleary and Brennan [21] replaced 2.5%, 5%, 7.5%, and 10% semolina with barley β-glucan fibre and found that pasta with 5%, 7.5%, and 10% exhibited lower reducing sugar than control where 2.5% shows the same reducing sugar release. Lu, Brennan, Serventi, Liu, Guan and Brennan [18] incorporated 5%, 10%, and 15% shiitake mushroom powder into pasta. The authors found that 10% and 15% incorporation with lower AUC but 5% incorporation shows the same AUC compared to control. Chusak et al. [22] had substituted semolina with gac fruit (*Momordica cochinchinensis*) powder and found a significantly decreased AUC when 10–15% semolina was replaced with fruit powder, while the same AUC compared to control was found with 5% semolina replacement. There are many studies, which substitute semolina (or wheat) with other materials, that show a reduced glycaemic response [17,23,24,25]. The lower AUC may be partially due to a reduction in overall starch content (Table 2) caused by semolina substitution. The correlation test (Appendix A) shows a positive correlation between starch content and AUC (*r*^2^ = 0.806, *p* ≤ 0.05 for spinach pasta; *r*^2^ = 0.829, *p* ≤ 0.05 for red cabbage pasta), indicating lower starch content of spinach and red cabbage pasta as factor leading to lower AUC.

### 3.2. In Vitro Starch Digestion Versus α-Amylase Inhibition Effect

Spinach juice 1% and spinach puree 2% pasta had lower GI than control, showing a decreased AUC (Figure 1c), possibly due to α-amylase inhibition of spinach, as shown in Table 3. The extracts from spinach show a higher α-amylase inhibition ability than red cabbage. The 50% inhibition concentration (IC_50_) was 0.047, 0.053, 0.240 mg/mL, for spinach juice, spinach puree and spinach pomace, respectively. Therefore, the pasta samples such as spinach juice 1% and spinach pomace 10% pasta also presented measurable α-amylase inhibition effect. Barkat et al. [26] have reported a high α-amylase inhibition ability (comparable to acarbose, a known α-amylase inhibitor) of spinach materials. The authors showed that the α-amylase inhibition effect of spinach was influenced by harvest day (weakest at 20 days, strongest at 60 days). Red cabbage extracts also showed some α-amylase inhibition ability, but their inhibition effect was much lower than spinach extracts. As the results show (Table 3), red cabbage pasta samples (RCJ1 and RCPO10) showed unmeasurable α-amylase inhibition ability. McDougall et al. [27] found that phenolic compounds extracts from red cabbage express no α-amylase inhibition at low dose (100 µg), and the inhibition effects rank the weakest at a higher dose (500 µg) compared with strawberry, blackcurrant, blueberry, raspberry, and green tea. Yusuf et al. [28] found that carrot varieties exhibit dramatically different α-amylase inhibition abilities. The α-amylase inhibition effect of spinach and red cabbage extracts may come from phenolic compounds. Barrett et al. [29] have shown that some phenols can interact with enzymes such as α-amylase, which results in blocking their catalytic sites, thus reducing their activity. Takahama and Hirota [30] showed that flavonoids in foods could interact with starch to form starch-flavonoid complexed through covalent bonds and hydrophobic interactions, thus helping to suppress amylose-iodine formation and further suppress amylose digestion through spectrophotography.

### 3.3. In Vitro Starch Digestion Versus Dietary Fibre

Spinach and red cabbage juice, puree and pomace contained more dietary fibre than semolina. Table 2 shows the dietary fibre profile of spinach and red cabbage pasta. Vegetable substitution increases the In vitro starch digestion versus dietary fibre total dietary fibre (TDF) content of all vegetable pasta samples compared to control (although some increases were not statistically significant). A substitution level of 10% pomace of spinach and red cabbage in this study increased the TDF, IDF and SDF the most dramatically. The dietary fibre of vegetable pasta is dependent on fibre from raw material and possible cooking loss. Sobota and Zarzycki [31] found that cooking impacts TDF, IDF and SDF of pasta product, and that impact depends on pasta type and cooking time. Appendix A shows the correlation factor between glycaemic response and dietary fibre. For spinach pasta, negative correlations were found between SDF vs. AUC (*r*^2^ = −0.763, *p* ≤ 0.05) and TDF vs. AUC (*r*^2^ = −0.756, *p* ≤ 0.05), indicating more soluble fibre and total fibre accompanied with spinach materials results in lower AUC. For red cabbage pasta, negative correlation was found between SDF vs. AUC (*r*^2^ = −0.758, *p* ≤ 0.05). Chau et al. [32] has illustrated that fibre can reduce post-prandial serum glucose via hindering the diffusion of glucose, retarding α-amylase action. Brennan and Tudorica [33] found that dietary fibre alter the water-binding activity of pasta samples, and restrict starch swelling during cooking influencing starch digestion. Peressini, Cavarape, Brennan, Gao and Brennan [17] has tested AUC of soluble fibre enriched pasta. Authors found a decreased AUC with Barley Balance^®^, psyllium but increased AUC for inulin and inulin HPX when 15% semolina was substituted by soluble fibre. Researchers suggested that changes in pasta structure induced by fibre enrichment contribute to the glycaemic response of pasta products. Although dietary fibre is believed to deliver health benefits such as preventing constipation, reducing bowel transit time, and selectively promoting of beneficial microbiota [34]. The dietary fibre can negatively affect the pasta quality due to the disruption of the gluten matrix [7], and lead to competition with starch to bind with protein [35] and water [36].

### 3.4. Total Phenolic Contents (TPC)

Total Phenolic Contents (TPC) results of raw, cooked, and digested pasta is shown in Table 4. A higher (*p* < 0.05) TPC value was found in all spinach and red cabbage fortified pasta samples compared to control for the raw pasta. As a result, cooked pasta performed similar trends. The 10% pomace substitution showed the highest TPC value for both spinach pasta and red cabbage pasta. At the same substitution level, juice samples showed the highest TPC value while pomace samples showed the lowest (spinach juice 1% > spinach puree 1% > spinach pomace 1%; red cabbage juice 1% > red cabbage puree 1% > red cabbage pomace 1%). This trend was observed in both raw and cooked pasta samples. Cooked pasta had a lower (*p* < 0.05) TPC value than raw pasta for all the pasta samples, indicating a loss during cooking. Fares et al. [37] reported that boiling water can degrade sensitive phenols while releasing some bond phenolics.

After digestion, the TPC value of all pasta samples increased dramatically compared to cooked pasta (spinach juice 1% increased from 317 to 1866). This trend was similar to the one report by Koehnlein et al. [38], who tested TPC value of most consumed foods of Brazilian diet before and after digestion. The authors found that TPC value of egg pasta increased 6 times after digestion. In cereals, phenolics can be conjugated to sugars, cell wall polysaccharides, or amines [38]. In vitro digestion can release those conjugated phenolics because starch and proteins, with which those phenols bond, are hydrolysed [39]. In this study, TPC value of control pasta increased from 232 mg GAE/100 g before digestion to 1725 mg GAE/100 g after digestion. It indicates semolina itself can release lots of bonded phenols after digestion. The spinach and red cabbage fortified pasta retained a higher (*p* < 0.05) TPC compared to control after in vitro digestion. This illustrates more bioavailable phenolics from spinach and red cabbage pasta than control pasta. At the same 1% substitution level, juice-fortified pasta has the highest bioavailable phenols compared to puree or pomace-fortified pasta (Table 4). It suggested that juice-fortified pasta has more bioavailable phenols than puree and pomace fortified one.

### 3.5. β-Carotene Content of Spinach Pasta and Raw Material

β-carotene is a common lipid soluble carotenoid found abundantly in spinach material [40]. Table 5 shows the β-carotene content of spinach raw material and spinach pasta (The chromatographic separation of β-carotene is shown in Appendix A). The spinach juice and spinach puree contained the same amount (*p* < 0.05) of β-carotene, while spinach pomace contained less β-carotene than spinach juice and puree (*p* < 0.05), potentially because pomace production involves more than 7 h of 60 °C air drying, which may cause loss of β-carotene. Hiranvarachat et al. [41] found that around 15% of β-carotene loss from blenched carrot samples after 7 h of hot air drying. The author contribute that loss to isomerization degradation, which transfers β-carotene to its cis form, and thermal degradation. Marx et al. [42] have found that blenching prevents nonenzymatic browning reaction and causes a higher value of β-carotene of pasteurised carrot juice. This may be another reason that spinach pomace has lower β-carotene as spinach pomace preparation was not involved in blench processing but was related to hot air drying.

As a result, spinach juice and puree pasta have higher β-carotene content when the substitution level is the same (spinach juice 1% = spinach puree 1% > spinach pomace 1% for uncooked pasta samples). It suggests that spinach juice pasta and spinach puree pasta have higher efficiency in delivering β-carotene than spinach pomace pasta. All spinach pasta samples contained higher β-carotene content than control, indicating that pasta is a compelling medium to deliver β-carotene. Compared to raw pasta, cooked pasta contained a comparable amount of β-carotene, which suggests that cooking does not cause a loss of β-carotene in spinach pasta, unlike for the total phenolic content loss observed during pasta cooking. It may be suggested that β-carotene is not water-soluble, so spinach pasta cooked in boiled water does not cause β-carotene loss to the cooking water. However, after in vitro digestion, the β-carotene was lower (*p* < 0.05) than cooked pasta samples, which showed the loss of β-carotene during digestion. Courraud et al. [43] reported a more than 50% β-carotene loss for raw and cooked spinach during digestion. The authors also showed less than 10% β-carotene loss during the digestion of carrot juice. It was hypothesized that the loss depends on the food matrix and raw material processing. For the pasta samples, digestion causes the decrease of β-carotene, the range of decrease was from 21.2% (control) to 25.9% (spinach puree 2%). After digestion, all spinach pasta samples contain more β-carotene than control, which illustrates spinach pasta can deliver significantly more bioavailable β-carotene than traditional durum pasta.

### 3.6. Antioxidant Capacities

The antioxidant capacities of foods depend on compounds such as phenolic content, vitamin C, vitamin E and carotenoids [44]. Antioxidant capacities, including Ferric Reducing Antioxidant Power (FRAP) and ABTS radical inhibition capacities of raw, cooked, and digested pasta, are shown in Figure 2. A higher FRAP capacity ABTS value was observed in all spinach and red cabbage pasta samples compared to control. A strong positive correlation was found (*r*^2^ = 0.915, *p* ≤ 0.01) between FRAPS values and ABTS values. Figure 2 illustrates that the incorporation of spinach and red cabbage material can improve the radical scavenging activities of pasta samples, indicating that pasta is a good medium to incorporate with vegetable material to improve human nutrition. At the same 1% substitution level, juice fortified pasta samples showed the highest antioxidant value compared to puree and pomace fortified pasta samples. This applied to red cabbage, and spinach fortified pasta on their raw, cooked and digested form. It may indicate that vegetable juice could be a solid choice to add to pasta formula to improve pasta antioxidant capacity compared to puree and pomace.

Raw and cooked red cabbage pasta samples exhibited higher antioxidant capacity than their spinach pasta counterparts. (e.g., red cabbage juice 1% raw higher than spinach juice 1% raw; red cabbage juice 1% cooked higher than spinach juice 1% cooked). However, after digestion, antioxidant capacity of spinach pasta increased while red cabbage pasta antioxidant capacity decreased. As a result, the FRAP value of digested red cabbage juice 1% was equal to (*p* < 0.05) of digested spinach juice 1%, and the ABTS value of digested red cabbage juice 1% was lower (*p* < 0.05) than digested spinach juice 1%. This may be due to the different antioxidant stability of spinach and red cabbage materials. Anti-oxidant capacities of red cabbage materials is mainly from its water-soluble anthocyanin [45]. McDougall et al. [46] reported that anthocyanin content from red cabbage was stable to gastric digestion but suffered a heavy loss (around 73%) during pancreatic digestion. Such changes may explain the decreased antioxidant capacity of red cabbage pasta after in vitro digestion. For control and spinach pasta, digested samples showed higher antioxidants than cooked pastas, potentially due to less loss of carotenoid (such as β-carotene) during digestion and conjugated phenols release after digestion. Those phenols express their additional antioxidant abilities.

## 4. Conclusions

The addition of vegetable ingredients was meaningful to enhance the nutritional value of pasta. In vitro starch digestion results suggested that the glycaemic response was modulated by a combination effect of the following several factors: starch and dietary fibre content, influenced by the amount of vegetables substituting the semolina. Another factor is the α-amylase inhibition effect of vegetable materials, which is the native properties of vegetable varieties. Results showed that spinach juice can reduce the GI of resultant pasta even at only 1% substitution. Antioxidant, total phenolic content, and β-carotene content results are encouraging because bioavailable phytochemicals increased after vegetable material was added to pasta formula. The bioavailable phytochemicals are dependent on vegetable varieties and their forms. Cooking influence on those phytochemicals was also present. It is plausible that cooking reduced water-soluble nutrients such as total phenolics but did not influence on β-carotene, which is lipid soluble, of the resultant pasta. The outcome of the study could be helpful for the food industry in designing nutritious vegetable pasta. Further study could be conducted to research bioaccessibility of those phytochemicals of the vegetable pasta.

## Figures and Tables

**Figure 1 nutrients-14-04575-f001:**
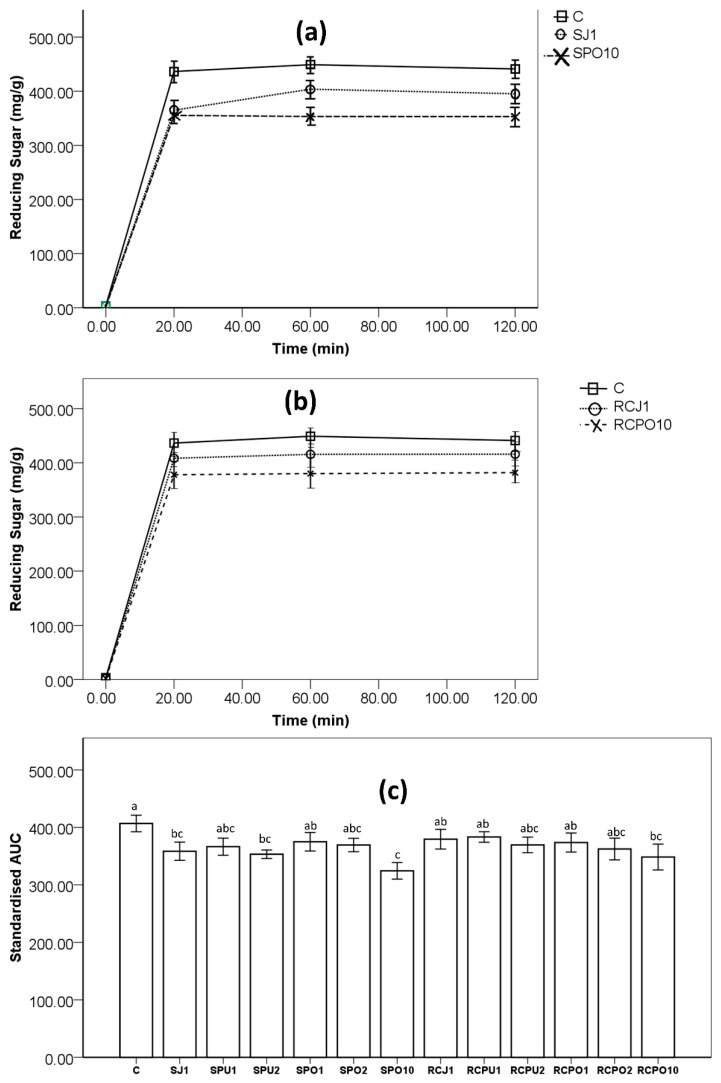
(**a**) reducing sugar release vs. time of C (control), spinach juice 1% pasta (SJ1), spinach pomace 10% pasta (SPO10); (**b**) reducing sugar release vs. time of of C (control), red cabbage juice 1% pasta (RCJ1), red cabbage pomace 10% pasta (RCPO10); (**c**) Value for are under the curve (AUC) of pasta samples, SJ, SPU, SPO represent spinach juice pasta, spinach puree pasta, spinach pomace pasta, respectively; RCJ, RCPU, RCPO represent red cabbage juice pasta, red cabbage puree pasta, red cabbage pomace pasta, respectively; C: control sample, 1, 2 and 10 is the substitution level (g/100 g or %) based on dry weight. The same letters are not significantly different from each other (*p* > 0.05) according to the ANOVA- Duncan test.

**Figure 2 nutrients-14-04575-f002:**
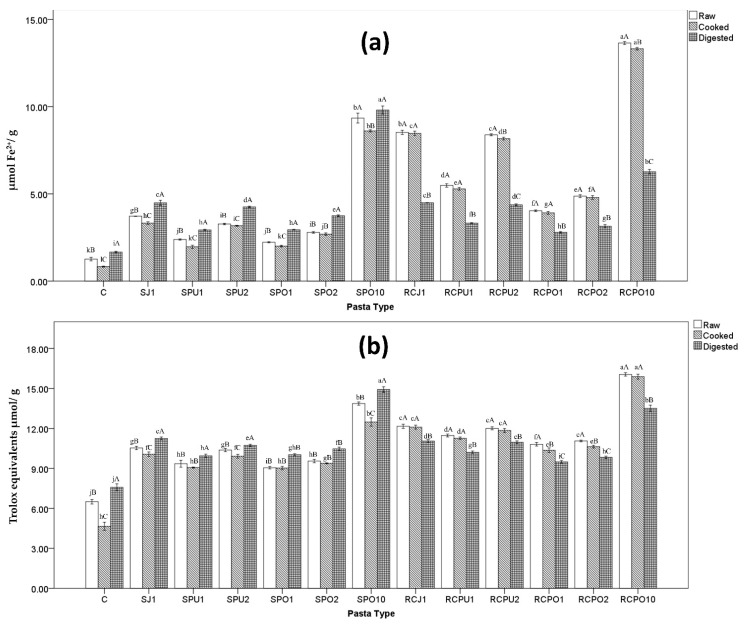
FRAP (**a**) and ABTS (**b**) value of vegetable pasta, SJ, SPU, SPO represent spinach juice pasta, spinach puree pasta, spinach pomace pasta, respectively; RCJ, RCPU, RCPO represent red cabbage juice pasta, red cabbage puree pasta, red cabbage pomace pasta, respectively; C: control sample, 1, 2 and 10 is the substitution level (g/100 g or %) based on dry weight. The same lowercase letters are not significantly different from each other (*p* > 0.05) for the same states of pasta (raw vs. raw, cooked vs. cooked, digested vs. digested). The same uppercase letters are not significantly different from each other (*p* > 0.05) for the same kind of pasta (raw vs. cooked vs. digested).

**Table 1 nutrients-14-04575-t001:** Pasta recipes (130 g batches).

Pasta Type	Semolina g	Water g	Vegetable Amount g	Dry Matter from Vegetable g
C	100	30	0	0
Spinach Pasta				
SJ1	99	21	10	1
SPU1	99	12	19	1
SPU2	98	23	9	2
SPO1	99	30	1	1 *
SPO2	98	30	2	2 *
SPO10	90	30	10	10 *
Red Cabbage Pasta				
RCJ1	99	24	7	1
RCPU1	99	23	9	1
RCPU2	98	23	9	2
RCPO1	99	30	1	1 *
RCPO2	98	30	2	2 *
RCPO10	90	30	10	10 *

SJ, SPU, SPO represent spinach juice pasta, spinach puree pasta, spinach pomace pasta, respectively; RCJ, RCPU, RCPO represent red cabbage juice pasta, red cabbage puree pasta, red cabbage pomace pasta, respectively; C: control sample, 1, 2 and 10 is the substitution level (g/100 g or %) based on dry weight. * the water content of pomace is neglected in this study because they are less than 14% which is close to semolina.

**Table 2 nutrients-14-04575-t002:** Dietary Fibre and Starch content of cooked pasta.

Pasta Type	Total Starch g/100 g	Total Dietary Fibre g/100 g	Soluble Dietary Fibre g/100 g	Insoluble Dietary Fibre g/100 g
Spinach Pasta				
C	70.2 ± 0.3 ^a^	3.72 ± 0.30 ^d^	1.46 ± 0.18 ^c^	2.26 ± 0.12 ^d^
SJ1	69.5 ± 0.5 ^b^	3.88 ± 0.28 ^cd^	1.58 ± 0.01 ^c^	2.30 ± 0.29 ^d^
SPU1	69.5 ± 0.5 ^b^	3.95 ± 0.07 ^cd^	1.58 ± 0.03 ^c^	2.37 ± 0.04 ^cd^
SPU2	68.4 ± 0.5 ^c^	4.26 ± 0.11 ^c^	1.63 ± 0.05 ^bc^	2.63 ± 0.06 ^bc^
SPO1	69.3 ± 0.4 ^b^	4.17 ± 0.09 ^c^	1.66 ± 0.07 ^bc^	2.51 ± 0.03 ^cd^
SPO2	68.6 ± 0.5 ^c^	4.74 ± 0.09 ^b^	1.89 ± 0.12 ^b^	2.93 ± 0.0 4 ^b^
SPO10	60.8 ± 0.36 ^d^	9.01 ± 0.11 ^a^	3.13 ± 0.02 ^a^	5.89 ± 0.13 ^a^
Red Cabbage Pasta				
C	70.2 ± 0.3 ^a^	3.72 ± 0.30 ^f^	1.46 ± 0.18 ^d^	2.26 ± 0.12 ^f^
RCJ1	68.4 ± 0.5 ^b^	4.04 ± 0.09 ^ef^	1.65 ± 0.18 ^bcd^	2.40 ± 0.15 ^ef^
RCPU1	68.5 ± 0.4 ^b^	4.16 ± 0.08 ^de^	1.57 ± 0.04 ^cd^	2.58 ± 0.04 ^de^
RCPU2	67.3 ± 0.6 ^c^	4.47 ± 0.14 ^cd^	1.77 ± 0.08 ^bc^	2.86 ± 0.05 ^c^
RCPO1	68.4 ± 0.4 ^b^	4.63± 0.04 ^c^	1.68 ± 0.11 ^bcd^	2.78 ± 0.03 ^cd^
RCPO2	67.3 ± 0.4 ^c^	5.29 ± 0.07 ^b^	1.87 ± 0.03 ^b^	3.43 ± 0.10 ^b^
RCPO10	60.0 ± 0.5 ^d^	12.07 ± 0.11 ^a^	3.57 ± 0.18 ^a^	8.51 ± 0.07 ^a^

SJ, SPU, SPO represent spinach juice pasta, spinach puree pasta, spinach pomace pasta, respectively; RCJ, RCPU, RCPO represent red cabbage juice pasta, red cabbage puree pasta, red cabbage pomace pasta, respectively; C: control sample, 1, 2 and 10 is the substitution level (g/100 g or %) based on dry weight. values within a column followed by the same letter are not significantly different from each other (*p* > 0.05) at the same pasta group, according to the ANOVA-Duncan test.

**Table 3 nutrients-14-04575-t003:** α-amylase Inhibition of Raw Materials and Spinach Pasta Sample.

Sample	Inhibition IC_50_ Value
Raw Material	
Spinach juice	0.05 mg/ml
Spinach puree	0.05 mg/ml
Spinach pomace	0.24 mg/ml
Red cabbage juice	4.59 mg/ml
Red cabbage puree	3.02 mg/ml
Red cabbage pomace	12.5 mg/ml
Pasta Sample	
SJ1	6.57 mg/ml
SPO10	3.12 mg/ml
RCJ1	>100 mg/ml
RCPO10	>100 mg/ml

SJ, SPO represent spinach juice pasta, spinach pomace pasta, respectively; RCJ, RCPO represent red cabbage juice pasta, red cabbage pomace pasta, respectively; 1, 10 is the substitution level (g/100 g or %) based on dry weight. Lower number indicates a higher inhibition effect.

**Table 4 nutrients-14-04575-t004:** Total Phenolic content and antioxidant capacity of spinach and red cabbage pasta.

	TPC (mg GAE/100 g)		
Pasta Type	Raw	Cooked	Digested
Spinach pasta			
C	441 ± 4 ^e^	232 ± 3 ^d^	1725 ± 33 ^d^
SJ1	571 ± 18 ^b^	317 ± 4 ^b^	1866 ± 14 ^b^
SPU1	490 ± 8 ^d^	273 ± 4 ^c^	1788 ± 18 ^c^
SPU2	522 ± 3 ^c^	327 ± 9 ^b^	1869 ± 35 ^b^
SPO1	456 ± 8 ^e^	266 ± 2 ^c^	1781 ± 21 ^c^
SPO2	499 ± 14 ^cd^	315 ± 6 ^b^	1854 ± 49 ^b^
SPO10	850 ± 29 ^a^	621 ± 14 ^a^	2023 ± 23 ^a^
Red cabbage pasta			
C	441 ± 4 ^g^	231 ± 3 ^f^	1725 ± 33 ^d^
RCJ1	804 ± 4 ^b^	532 ± 4 ^b^	1934 ± 26 ^b^
RCPU1	713 ± 10 ^d^	367 ± 2 ^d^	1897 ± 26 ^bc^
RCPU2	779 ± 7 ^c^	497 ± 11 ^c^	1922 ± 43 ^b^
RCPO1	549 ± 3 ^f^	285 ± 5 ^e^	1832 ± 29 ^c^
RCPO2	600 ± 5 ^e^	369 ± 5 ^d^	1894 ± 67 ^bc^
RCPO10	1140 ± 7 ^a^	791 ± 8 ^a^	2070 ± 27 ^a^

SJ, SPU, SPO represent spinach juice pasta, spinach puree pasta, spinach pomace pasta, respectively; RCJ, RCPU, RCPO represent red cabbage juice pasta, red cabbage puree pasta, red cabbage pomace pasta, respectively; C: control sample, 1, 2 and 10 is the substitution level (g/100 g or %) based on dry weight. values within a column followed by the same letter are not significantly different from each other (*p* > 0.05) at the same pasta group, according to the ANOVA- Duncan test.

**Table 5 nutrients-14-04575-t005:** β-carotene content of spinach raw material and pasta.

Spinach Raw Material μg/g			
Spinach juice	27.6 ± 0.8 ^a^		
Spinach puree	28.2 ± 0.8 ^a^		
Spinach pomace	17.9 ± 0.6 ^b^		
**Spinach Pasta**			
**Pasta Type**	**Raw μg/g**	**Cooked μg/g**	**Digested μg/g**
C	0.03 ± 0.00 ^fA^	0.03 ± 0.00 ^eA^	0.03 ± 0.00 ^fB^
SJ1	0.32 ± 0.01 ^dA^	0.31 ± 0.01 ^cA^	0.24 ± 0.01 ^dB^
SPU1	0.31 ± 0.01 ^dA^	0.30 ± 0.01 ^cA^	0.23 ± 0.01 ^dB^
SPO1	0.19 ± 0.01 ^eA^	0.18 ± 0.01 ^dA^	0.14 ± 0.01 ^eB^
SPU2	0.58 ± 0.01 ^bA^	0.57 ± 0.01 ^bA^	0.42 ± 0.01 ^bB^
SPO2	0.36 ± 0.01 ^cA^	0.36 ± 0.01 ^cA^	0.27 ± 0.01 ^cB^
SPO10	1.99 ± 0.01 ^aA^	2.00 ± 0.15 ^aA^	1.50 ± 0.03 ^aB^

C—control. SJ1—Spinach juice 1%; SPU1—Spinach Puree 1% pasta; SPO1—Spinach pomace 1% pasta; SPU2—Spinach puree 2% pasta, SPO2—Spinach pomace 2% pasta; SPO10—spinach pomace 10% pasta; The same lowercase superscripted letters within a row are not significantly different from each other (*p* > 0.05), according to the ANOVA-Duncan test. The same uppercase superscripted letters within a column are not significantly different from each other (*p* > 0.05).

## Data Availability

Not applicable.

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
