# Peer review of "Predictive Glycaemic Response of Pasta Enriched with Juice, Puree, and Pomace from Red Cabbage and Spinach"

_nutrients, 2022, doi:10.3390/nu14214575_

Round 1

Reviewer 1 Report

2.5- In vitro digestion is not described in detail what equipment was used. 

2.8-Determination of alpha-amylase  inhibition - uses the word: extract. Is the same extract that supernatant in 2.6

2.10-testing of B-carotene was only a duplicate. It should be three independent replicate.

The relevance of using the biomarkers selected to support the soundness of the paper is not explained properly in the introduction or the conclusions. 

Reviewer 2 Report

1. The title of the manuscript reflects predictive response, but the authors have not developed any predictive models in the study for fortification study. The authors should incorporate any predictive model to address their claim.

2. The HPLC chromatograms of beta carotene analysis in this study should be included in supplementary data.

3. What is the novelty of the study conducted by the authors? Authors should explain about the novelty of the study with current state art in the fortification field.

Reviewer 3 Report

The reviewed manuscript contains the results of an interestingly designed experiment, in which pasta enriched with purees, juices and pomace from spinach and red cabbage were studied.  The purpose of the experiment was to compare the nutritional value of the resulting pastas. Both the introduction to the research problem and the objectives of the project were presented in a clear and essential manner, sufficient for the needs of the article submitted for review.  The authors showed that the addition of vegetables, in the form of various forms, had a significant effect on the increase in the nutritional value of the macarons studied.

I find this manuscript valuable and interesting due to the cognitive and application values it brings, but nevertheless I have a few minor comments on it.

Line 57-61: It might be worthwhile to cite the unpublished results of the preliminary studies reported here in the form of tables or figures in the supplementary materials. 

Line 71: What guided the choice of temperature for drying pomace? Wasn't it taken into account that high drying temperature of spinach has an effect on the decrease of β-carotene concentration?

According to Lefsrud et al. (2008), when the drying temperature of spinach was increased from -250C (freeze-drying) to 750C (oven drying), the concentration of β-carotene decreased over 70%.   

Lefsrud, M., Kopsell, D., Sams, C., Wills, J., & Both, A. J. (2008). Dry matter content and stability of carotenoids in kale and spinach during drying. HortScience43(6), 1731-1736.

Line 394-396: Delete

Line 397-398: Supplementary Materials: instead of Figure S1: title; Table S1: title; Video S1: title, should be Table S1 and S2 with their actual titles.

Supplementary materials: Table 1 should have a more precise title, referring to what it represents. No title for Table 2. Both Table 1 and Table 2 lack explanations of the abbreviations used in them.

Round 2

Reviewer 1 Report

Changes were reviewed and addressed comments

Author Response

Thanks for your review

Reviewer 2 Report

The HPLC chromatogram pattern of beta carotene present in fortified food products are very much essential for publishing this study, since it is the only instrumental data in this study to improve its research credibility. The authors should compare the HPLC chromatogram of standard beta carotene with chromatogram of food samples to give reliability to the data provided by them.

Author Response

Thanks for your suggestion, I have uploaded supplementary Figure S1 as an example chromatogram of beta-carotene (digested spinach pomace 10%) and mentioned it in the article accordingly.